# Effects of Time-Restricted Feeding on Supramaximal Exercise Performance and Body Composition: A Randomized and Counterbalanced Crossover Study in Healthy Men

**DOI:** 10.3390/ijerph18147227

**Published:** 2021-07-06

**Authors:** Joana M. Correia, Inês Santos, Pedro Pezarat-Correia, Cláudia Minderico, Brad J. Schoenfeld, Goncalo V. Mendonca

**Affiliations:** 1Laboratório de Função Neuromuscular, Faculdade de Motricidade Humana, Universidade de Lisboa, Estrada da Costa, 1499-002 Cruz Quebrada-Dafundo, Portugal; ppezarat@fmh.ulisboa.pt (P.P.-C.); gvmendonca@gmail.com (G.V.M.); 2CIPER—Centro Interdisciplinar Para o Estudo da Performance Humana, Faculdade de Motricidade Humana, Universidade de Lisboa, Estrada da Costa, 1499-002 Cruz Quebrada-Dafundo, Portugal; cminderico@gmail.com; 3CIDEFES—Centro de Investigação em Desporto, Educação Física, Exercício e Saúde, Universidade Lusófona, Campo Grande 376, 1749-024 Lisboa, Portugal; isantos@fmh.ulisboa.pt; 4Laboratório de Nutrição, Faculdade de Medicina, Universidade de Lisboa, Av. Prof. Egas Moniz, 1649-028 Lisboa, Portugal; 5Department of Health Sciences, CUNY Lehman College, Bronx, NY 10468, USA; brad.schoenfeld@lehman.cuny.edu

**Keywords:** intermittent fasting, Wingate test, body composition, training

## Abstract

Using a crossover design, we explored the effects of both short- and long-term time-restricted feeding (TRF) vs. regular diet on Wingate (WnT) performance and body composition in well-trained young men. Twelve healthy male physical education students were included (age: 22.4 ± 2.8 years, height: 174.0 ± 7.1 cm, body mass: 73.6 ± 9.5 kg, body mass index: 24.2 ± 2.0 kg/m^2^). The order of dieting was randomized and counterbalanced, and all participants served as their own controls. TRF was limited to an 8-h eating window and non-TRF involved a customary meal pattern. Participants performed WnT tests and body composition scans at baseline, post-one and post-four weeks of the assigned diet. Before testing, participants were asked to fill out a dietary record over four consecutive days and were instructed to continue their habitual training throughout the study. Energy intake and macronutrient distribution were similar at baseline in both conditions. WnT mean power and total work output increased post-four weeks of TRF. Both conditions were similarly effective in increasing fat-free mass after four weeks of intervention. However, there was no correlation between change in fat-free mass and WnT mean power after TRF. TRF did not elicit any changes in WnT performance or body composition one week post-intervention. Thus, long-term TRF can be used in combination with regular training to improve supramaximal exercise performance in well-trained men.

## 1. Background

Fasting is characterized by the voluntary abstinence from food intake for a specified time window in a day, with or without caloric restriction (CR) [1]. This dietary approach is distinct from CR, in which the daily caloric intake is reduced chronically by 15 to 40% in most individuals, with maintenance of meal frequency [2]. Intermittent fasting (IF) is a term that encompasses many definitions spanning from absolute or partial restriction in energy intake (i.e., 50 and 100 % restriction of total daily energy intake) to ad libitum intake. It may be prescribed for varying periods (usually for 12 h or more) during the day and it may or may not involve overnight food abstinence (i.e., non-Ramadan vs. Ramadan IF) [3]. In recent years, a focus on several forms of IF has emerged in the scientific literature (i.e., alternate-day fasting, intermittent energy restriction, and time-restricted feeding) [4,5]. Time-restricted feeding (TRF) is a form of IF that recently gained popularity because it encompasses a cyclic alternation between feasting (ad libitum energy intake) and fasting on a daily basis (from 12–21 h per day) [3,6].

Many athletes voluntarily restrict their energy intake to achieve a specific body mass category, for aesthetic reasons, or to attain a better strength-mass ratio to improve performance [7]. Training and IF are two effective strategies for reducing body fat via heightened lipolysis in adipose tissue [8]. This is important for athletes, as they need to control their body composition to optimize the balance between fat mass (FM) and fat-free mass (FFM) to achieve high levels of motor performance [3,5]. Performance in Wingate (WnT) testing has been shown to be highly dependent on energy release from both anaerobic and aerobic processes [9,10]. Evidence also indicates that the relative contribution of anaerobic and aerobic energy supply for WnT power output varies as a function of sports specialization [9] and time of day [11]. In addition to aerobic endurance and anaerobic capacity, a high power-to-weight ratio is important during WnT [12]. A high level of WnT performance is desirable because it predicts realistic times at 100, 200, and 400 m running distances [13]. In female athletes, it is also related to performance in running the 800 m, but not in longer distances [14]. Unfortunately, the chronic impact of IF on WnT performance has only been studied within the context of CR and Ramadan [15,16]. In both circumstances, IF was shown to be generally detrimental for WnT performance [13,14,15,16]. Despite being pertinent, these findings cannot be extrapolated to alternative IF approaches, such as those not involving an obligatory CR or coercive dehydration. This is important because IF has been shown to be intimately associated with several positive physiological adaptations (e.g., [17,18]), independent of CR [17]. In addition, recent meta-analytic data clearly shows the differential impact of Ramadan vs. non-Ramadan IF on exercise performance [19]. Thus, our main purpose was to investigate the effects of long-term TRF without CR on WnT performance of well-trained, physically active healthy male physical education students. In addition, we sought to determine whether the effects of TRF on WnT performance are associated with alterations in FM and FFM. We hypothesized that the TRF regimen would lead to greater fat loss and improvements in anaerobic performance as compared to a regular dietary pattern. 

## 2. Methodology

### 2.1. Participants

Eighteen male physical education students, with normal blood pressure (all ≤ 120/80 mmHg) and not taking any medication, started this crossover trial [20]. Six of them were excluded during the first 2 weeks owing to lack of adherence to the study conditions. Therefore, twelve participants finished the study (age: 22.4 ± 2.8 years, height: 174.0 ± 7.1 cm, body mass: 73.6 ± 9.5 kg, body mass index: 24.2 ± 2.0 kg/m^2^). All participants were similarly active, accumulating 9 h of physical activity per week as part of their academic work. Inclusion was limited to male participants, aged 18-30 years, exhibiting more than 3 yrs of continued experience and current participation in power-sports training (training frequency of at least 3 times per week). Exclusionary criteria included active smoking status, known metabolic disease, cardiovascular disease, respiratory disorders, and orthopedic issues limiting exercise performance. Participants received verbal information about the risks, requirements, and procedures of the study, and provided written informed consent prior to study entry, which was approved by the faculty ethics committee (CEFMH Nº12/2018) and conducted in accordance with the Declaration of Helsinki. Height and blood pressure (Tango SunTech Medical Morrisville, NC) were taken at baseline. All participants were well accustomed to WnT testing procedures (as part of their academic work and previous participation in studies involving this specific test). 

### 2.2. Experimental Design

Figure 1 depicts the experimental design of the present study. Participants were evaluated over the course of several different visits at approximately the same time of the day (between 06.00 and 08.00 a.m.). Measurements were taken before and after four weeks of each dietary intervention (TRF vs. non-TRF) and participants served as their own controls. The order of dieting was randomized and counterbalanced (i.e., at study entry, nine participants were randomly allocated to the TRF and another nine to the non-TRF dietary intervention). Minimization was used for the allocation sequence to minimize the imbalance in the number of participants at baseline for each group [21]. Among the six participants that were excluded due to lack of adherence, four had been previously assigned to begin with the non-TRF condition and two with the TRF condition. Thus, five participants began with the non-TRF intervention and seven with the TRF intervention. Finally, to examine the short-term effects of TRF, all participants were also tested post-1 week of TRF. Prior to each dietary intervention, participants were asked to fill out a dietary record over four consecutive days. They were also asked to continue their habitual training throughout the study (i.e., exercise routine, order of exercises, number of exercises, sets, repetitions, loads, interset time intervals, weekly frequency, and recovery days between exercise sessions). As in previous studies, a two weeks washout period (no specific diet or structured exercise) separated both conditions [22]. 

Within both conditions, and at each time point, testing involved two consecutive visits to the laboratory. During the first visit, height and body mass measurements were taken with the participants wearing lightweight clothes and no shoes. Subsequently, participants were scanned with dual-energy X-ray absorptiometry (DXA) to estimate body composition. On the second visit, participants performed a brief warm-up followed by a WnT test on a mechanically braked cycle ergometer. Each participant was instructed to avoid heavy exercise for at least 24 h before testing and to fast from 8 p.m. until the subsequent morning testing sessions. Participants abstained from consuming liquids from 8 p.m. until the first testing visit on the subsequent morning. In contrast, water was consumed ad libitum during the 24 h preceding the second visit to the laboratory. Participants were also asked to empty their bladders just before testing on both visits.

The experimental design consisted of two dietary interventions involving four weeks of time-restricted feeding (TRF) and non-time-restricted feeding (non-TRF). TRF interventions followed a 16/8 time-restricted feeding protocol. The order of dieting was randomized and counterbalanced and two weeks of washout separated both conditions. Baseline dietary records were obtained prior to each dietary intervention. Body composition and Wingate performance were determined in consecutive days at baseline and after four weeks of intervention in both conditions. In addition, these measures were taken after one week of TRF to characterize the short-term effects of this specific intervention. Participants were also asked to continue their habitual training throughout the study.

### 2.3. Wingate Anaerobic Test

The WnT protocol was conducted using a computerized cycle ergometer (Monark 824E Peak Bike, Vargerg Sweedenbic, Sweden). The cycle ergometer was equipped with toe-clips to prevent the participants’ feet from slipping. To avoid intraindividual postural changes between testing sessions, foot position on the pedals, saddle height, and upper-body position were adjusted to each participant’s anthropometric specifications and maintained identical throughout the study. Before testing, participants underwent a 5-min warm-up involving pedaling at a cadence of 50 rpm at a constant power output set at 100 W. The warm-up included two or three brief (3–5 s) sprinting bouts to a higher cycling speed (120 rpm) at a higher power output [23]. The intensity of the warm-up was chosen to increase the heart rate to approximately 140–150 beats/min [24]. After a 1-min rest, participants were instructed to pedal at full speed with the cycle ergometer unloaded for 5–8 s. At this stage, the full braking force (7.5% of participant’s body mass (kg)) was applied and a 30-s count was implemented [25]. During the test, participants remained seated and were verbally encouraged to pedal as fast as possible. The highest mechanical power (peak power) attained over 1 s during the course of the exercise and the mean power corresponding to the average power output values obtained during the 30 s were recorded and stored for subsequent analysis. The fatigue index was calculated as the percentage of decrease in power output throughout the test [26]. Testing was carried out in a laboratory at an environmental temperature between 21 and 24 °C and a relative humidity between 44 and 56%. Retest reliability and validity of power indices obtained during WnT testing have been well documented since the end of 1980 [10]. Nevertheless, more recent analyses revealed that the coefficient of variation for peak power, mean power, fatigue index, and total work output is 0.97, 1.54, 1.89, and 1.74%, respectively. In addition, the intraclass correlation coefficients obtained for these respective variables was 0.987, 0.984, 0.948, and 0.968 [27].

### 2.4. Total Work Output

Total work output (W_tot_) was computed as the sum of total energy demand (E_tot_) and kinetic energy of the flywheel at the 30th s of exercise (E_k_), as follows [9]:W_tot_ = E_tot_ + E_k_(1)

The E_tot_ for the work performed during the 30-s test was calculated by the following equation:E_tot_ (J) = 6.12 × load (kg) × 9.81 × n(2)
where n is the number of pedal revolutions in 30 s.

The E_k_ at the 30th s was given by the following equation for a Monark ergometer (52 teeth for the chain wheel and 14 teeth for the gear) with a flywheel inertia estimated at 0.38 kg·m^−2^:E_k_ (J) = 103 × *v*^2^(3)
where *v* is the pedal frequency in revolutions per second.

### 2.5. Body Composition

Body mass was measured using a calibrated digital scale (TANITA^®^ BF-350 body composition analyzer, Arlington Heights, IL, USA). Height was measured only at baseline using a stadiometer (standing digital scale/height rod attached). Body mass index was then calculated by dividing the participants’ mass in kilograms by the square of their height in meters. We used DXA to estimate % body fat (BF), FFM, and FM. Testing was performed with a total body scanner (fan-beam mode, software version 5.67, enhanced whole-body analysis, Hologic Explorer-W, Waltham, MA, USA). Based on test-retest measures including 10 participants (other than the ones included in this study), the coefficient of variation for FM and FFM in our laboratory is 1.7 and 0.8%, respectively. The intraclass correlation coefficients obtained for these respective variables were 0.997 and 0.999 [28].

### 2.6. Dietary Intake and Protocols

One week before testing (pre-TRF and non-TRF), participants were asked to fill out a dietary record over four consecutive days to estimate energy intake and macronutrient distribution. Participants were given specific instructions regarding how to estimate portion sizes and identify all food and fluid intake from a specialized manual. Food models were viewed by each participant to enhance precision. Total energy intake, carbohydrate, lipid, and protein content were calculated with Food Processor^®^ Nutrition Analysis software (ESHA, Salem, Oregon). Following the collection of dietary data, participants were instructed to maintain their habitual food preferences over the course of the study to minimize noncompliance. TRF interventions followed a 16/8 time-restricted feeding protocol [3]. All participants consumed two to three meals of ad libitum food intake during an 8-h period (between 1 and 9 p.m.). The remaining 16 hrs per 24-h time period constituted the fasting period during which participants only were allowed to consume water, tea, and coffee (without caloric additives). The non-TRF diet corresponded to the participants’ usual dietary pattern without any timing restrictions. 

### 2.7. Statistical Analysis

Before comparing both conditions (TRF vs. non-TRF), data were tested for normality and homoscedasticity with the Kolmogorov–Smirnov and Mauchly’s test, respectively. An independent researcher in our laboratory, who was blinded to treatment allocation, analyzed all data. Paired samples *t*-tests were used to compare baseline energy intake and macronutrient distribution at the beginning of each dietary regimen (i.e., immediately before four weeks of TRF and non-TRF). The short-term impact of TRF (i.e., one week) on WnT performance and body composition was also determined using paired samples *t*-tests. A 2-way ANOVA [condition (TRF vs. non-TRF) by time (pre- vs. post-dietary intervention)] with repeated measures was conducted on all dependent variables to determine the effects of TRF on WnT performance, as well as on body composition. To explore the role of improved body composition on WnT performance, we also examined the Pearson correlation coefficients between delta mean power and delta FFM from pre- to post-TRF time point. Finally, on an individual basis, we calculated the post-intervention magnitude of improvement in time required to complete the total amount of work produced during WnT at baseline. For instance, if, in response to 30 s of WnT, a given participant achieved a total work output of 17,500 (post-intervention) vs. 16,000 J (pre-intervention), then the magnitude of improvement equates to 1500 J. This implies that, at the post-intervention time point, this specific participant was able to complete the 16,000 J of work in less than 30 s. The time difference required to complete this specific amount of work between time points (pre- vs. post-intervention) was calculated to display changes in supramaximal exercise performance in units of time. The partial eta-squared values (proportion of total variance that is attributable to an effect) are reported to indicate effect sizes (ES) for significant findings. All data are reported as mean ± SD. Statistical significance was set at *p* < 0.05. Data analysis was carried out using Statistical Package for the Social Sciences (version 25.0, SPSS Inc., Chicago, IL, USA). 

## 3. Results

As shown in Table 1, energy intake and macronutrient distribution were similar at baseline between conditions (*p* > 0.05). 

### 3.1. Impact of TRF after One Week of Intervention

Table 2 summarizes the impact of TRF on body composition and WnT performance after 1 week of intervention. As can be seen, short-term TRF was not effective in eliciting any changes in body mass or body composition (*p* > 0.05). Similarly, WnT performance was not affected by this specific short-term intervention (*p* > 0.05). 

### 3.2. Impact of TRF after Four Weeks of Intervention

Table 3 depicts the effects of TRF and non-TRF on WnT performance. With the exception of that seen in FFM, no other effects (condition, time, or interaction) were detected for body composition. Both dietary protocols were equally effective in increasing FFM after 4 weeks of intervention (time main effect: F = 5.8, *p* = 0.035; ES = 0.34). 

We obtained a significant condition x time interaction for WnT absolute mean power (F = 5.6, *p* = 0.037; ES = 0.34) and total work (F = 5.6, *p* = 0.038; ES = 0.32). As depicted in Figure 2A,C, absolute mean power increased from pre- to post-4 weeks of TRF (*p* = 0.04), but no significant changes were observed for the non-TRF condition. Similar findings were also obtained for total work. WnT relative mean power also exhibited a trend toward a differential response between conditions (condition x time interaction: F = 4.5, *p* = 0.058; ES: 0.29) (Figure 2B,D). In contrast, neither condition led to any change in WnT peak power (absolute or relative) or fatigue index after 4 weeks of intervention (Table 3). 

We also endeavored to explore whether the improvements in WnT performance after four weeks of TRF might be associated with improved body composition. Because there was an increase in FFM and WnT absolute mean power post-TRF, we reduced these differences between time points to a single component of Δ FFM and of Δ WnT mean power, which were then used for correlation analysis. We found no significant correlation between the changes in FFM and WnT absolute mean power induced by four weeks of TRF (*r* = 0.27, *p* = 0.39). Finally, since the participants’ total work output increased significantly after TRF, we calculated their magnitude of improvement in time required to complete WnT total work obtained at baseline. These calculations indicate that, after TRF, participants improved their total work time by −1.1 ± 2.0 s. 

## 4. Discussion

Our primary findings can be summarized as follows. First, contrasting to that typically reported after Ramadan, we found that WnT performance of healthy well-trained young men can be improved post-four weeks of TRF (vs. non-TRF). Second, we also observed that short-term TRF does not improve WnT performance or body composition in this specific population. Third, the magnitude of work-time improvement resulting from four weeks of TRF was greater than 1 s and this is quite meaningful from an athletic perspective. 

Most studies focusing on the interaction between IF and motor performance have generally used training paradigms involving endurance exercise [29,30] or examined changes in physiological capacity elicited by Ramadan fasting [31]. Various factors related to endurance-exercise fatigue (e.g., depletion of carbohydrate stores, increased body temperature, and dehydration) are less likely to affect motor performance during physically demanding tasks lasting only a few seconds or a few minutes [32]. Moreover, the negative impact of Ramadan fasting on WnT performance is not relevant to TRF because most people can stay well-hydrated by unrestricted non-caloric fluid ingestion [16]. By showing that WnT performance improved after four weeks of TRF (vs. non-TRF) in healthy well-trained young men, our findings are well aligned with this concept. We also found that the mechanistic basis of heightened WnT performance does not depend on altered body composition post-TRF. Finally, our data indicate that short-term TRF (i.e., one week) is ineffective for changing individual WnT performance and body composition.

WnT power output is related to athletic performance in sports involving short bouts of supramaximal exercise (e.g., football, tennis, basketball, field hockey, running, rowing, canoeing, swimming, cross training, and wrestling) [14,16,33,34,35]. Power output is particularly critical for the fast speeds observed in mid-distance events [36]. We found that, after four weeks of TRF, participants improved their total work time by more than 1 s. This magnitude of improvement can be practically meaningful for track and field athletes who compete in running distances of 400/800 m, and may represent the difference between qualifying for a competition or even winning a race (e.g., the difference between the former and new 400 m running distance world record is of 15 centesimal seconds) [37]. This is even more relevant when considering that participants who enrolled in this investigation continued their habitual training throughout the study and exhibited no differences in daily dietary intake before each intervention (TRF vs. non-TRF). Ultimately, the result indicates that enhanced WnT performance was primarily linked with the physiological adaptations elicited by TRF per se. However, given that WnT performance did not improve after one week of intervention, it should also be emphasized that the magnitude of interaction between TRF and WnT performance is strongly mediated by the duration of TRF. 

The impact of fasting on WnT performance remains poorly understood. According to past research, while short-term TRF has detrimental effects on WnT peak power during the early phase of restriction, performance returns to that seen at baseline after 4 days of dieting [15]. Unfortunately, it is difficult to compare our results with those of Naharudin et al. (2018) because their experimental design involved a TRF duration that was limited to 10 days, involved a 40% daily CR, and only reported measures of peak power. Nevertheless, as we also did not observe any impact of short- or long-term TRF (i.e., one or four weeks, respectively) on peak power, it may be concluded that this specific marker of motor performance remains largely unaffected beyond one week of TRF and that this occurs similarly either with or without CR. In accordance, it can be concluded that short-term TRF exerts no significant impact on the ability of the lower-limb muscles to produce high mechanical power [10]. Conversely, four weeks of TRF led to a positive adaptation in WnT mean power, implying that this nutritional regimen is effective for improving muscular endurance of the lower limbs (i.e., their ability to sustain extremely high power outputs) [10]. The etiological basis of heightened mean power may relate to increases in the expression of SIRT1 and phosphorylated AMPK with exercise in the fasted state [38]. AMPK has numerous downstream effects on gene expression and is involved in the regulation of mitochondrial biogenesis, in the use of metabolic substrates, and autophagy [8]. On the other hand, SIRT1 is involved in the regulation of metabolic processes primarily related to mitochondrial adaptation via increased peroxisome proliferator-activated receptor gamma coactivator 1-alpha (PGC-1α) and SIRT1, suppressing apoptosis and oxidative stress [17,39]. Given that performance in WnT has been shown to be highly dependent on energy release from both anaerobic and aerobic processes, improved oxidative capacity (e.g., via mitochondrial biogenesis) resulting from 4 weeks of TRF potentially provides a partial explanation for our findings [9,10]. 

TRF was as effective as the control condition in increasing FFM after four weeks of intervention. Even though we did not control caloric intake during TRF, participants were asked to maintain their habitual food preferences over the course of the study and no weight loss was observed after four weeks of intervention. Therefore, our findings further substantiate that long-term TRF does not negatively impact FFM compared to traditional meal timing given a relatively high protein intake (1.9 g.kg^−1^.d^−1^). Indeed, increases in FFM post-TRF have been reported in two previous studies in which caloric intake was carefully controlled to avoid weight loss [40,41]. Moreover, TRF may have an adjunctive role in preserving or delaying FFM losses when combined with resistance training [3,5]. 

Although significant decreases in FM with TRF have been reported in some previous studies [2,3,6], we were unable to replicate such findings. There are many factors that likely explain the lack of reduction in FM with TRF. For one, the magnitude of the caloric deficit elicited by TRF in this study was not sufficient to reduce body fatness to a significant level. Yet, it should be noted that the participants did lose a small percentage of body fat post-TRF (Table 3). Alternatively, the effectiveness of TRF in reducing FM may vary as a function of baseline body fatness. This is supported by past reports showing that FM is more easily lost in individuals with overweight/obesity than in those exhibiting a healthy body composition [8].

This study has several limitations that should be taken into account. First, energy and macronutrient intake were estimated based on self-report using dietary record over four consecutive days, and this approach has known limitations for accurately assessing nutritional intake [3,41]. Second, our sample size was relatively small, somewhat compromising statistical power. However, the effect size of TRF on WnT mean power was large (partial eta square > 0.14), indicating a large proportion of the variance in the dependent variable was explained by the independent variable. Third, inclusion was limited to young, well-trained male physical education students and thus our findings cannot be generalized to elite athletes, female athletes, or older participants. Fourth, during the free-living period (washout before crossing over to the alternate condition (TRF or non-TRF)), participants were instructed to follow their regular dietary habits. This lack of standardization may have influenced the baseline results and lessened our ability to detect differences after one week of TRF. Finally, although efforts were made to ensure that the participants followed their habitual training practices throughout the study (i.e., through detailed instruction and weekly follow-up contact), the exercise sessions were not directly supervised by any member of the research team. That said, the crossover design would seemingly account for individual variability in this regard.

## 5. Conclusions

In conclusion, our results indicate that adherence to four weeks of a modified IF regimen, such as the one employed in this study (i.e., TRF with 16 h of fasting and 8 h of feasting), can be used in combination with regular training to improve muscle performance during supramaximal cycling (WnT performance) in young, well-trained men. Our findings support the hypothesis that TRF triggers a hormesis-like effect in the capacity to generate supramaximal power, which may be of practical relevance to athletes. However, no adaptation occurred after one week of TRF. Thus, when aiming at improving supramaximal motor performance with TRF, emphasis should be placed on prescribing diet interventions > one week. 

## Figures and Tables

**Figure 1 ijerph-18-07227-f001:**
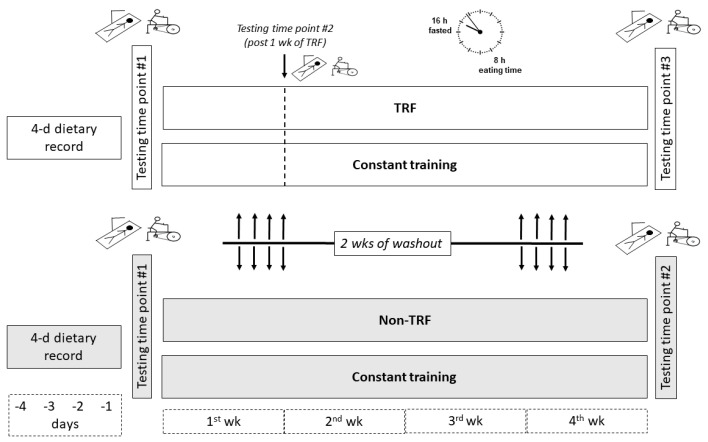
Schematic representation of the study protocol.

**Figure 2 ijerph-18-07227-f002:**
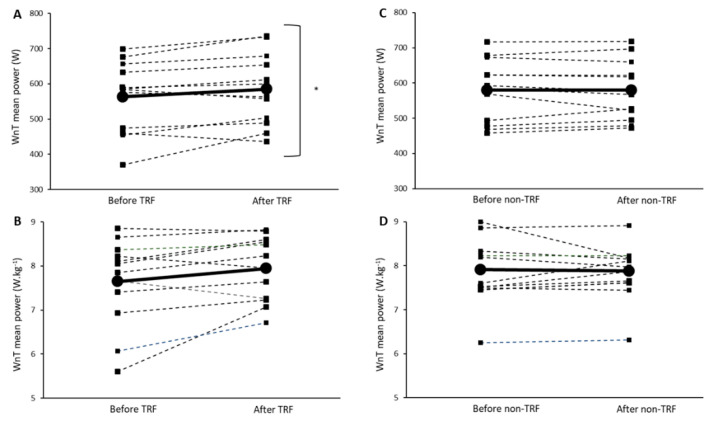
Wingate (WnT) mean power output obtained in response to 4 weeks of time-restricted feeding (TRF) and non-time-restricted feeding (non-TRF). Absolute (**A**) and relative (**B**) mean power in response to TRF. Absolute (**C**) and relative (**D**) mean power in response to non-TRF. Relative values were computed by dividing absolute power by the individual body mass. Values are mean ± SD. Dashed lines represent individual participants. The solid black line represents the mean value of all cases. * Significant differences between time points (*p* < 0.05).

**Table 1 ijerph-18-07227-t001:** Estimated daily dietary intake obtained four days before time-restricted feeding (TRF) and non-time-restricted feeding (non-TRF).

Variables	Before TRF	Before Non-TRF	*p* Value
Energy intake (kcal)	2655.7 ± 773.5 (2164.3–3147.2)	2576.7 ± 711.4 (2125.7–3028.7)	0.75
Carbohydrate (%)	45.9 ± 6.9 (41.5–50.2)	45.0 ± 4.5 (42.1–47.8)	0.73
Fat (%)	31.9 ± 4.8 (28.8–34.8)	31.3 ± 5.3 (27.8–34.6)	0.74
Protein (%)	20.3 ± 4.1 (17.7–22.8)	22.7 ± 3.0 (20.8–24.6)	0.10

Values are mean ± SD and 95% confidence interval.

**Table 2 ijerph-18-07227-t002:** Body composition and Wingate-derived variables obtained before and after one week of time-restricted feeding (TRF).

Variables	Pre-TRF	Post-1 Week TRF	*p* Value
*Body composition*			
Body mass	73.6 ± 9.5 (67.5–79.5)	73.5 ± 9.2 (67.6–79.4)	0.79
Fat mass (kg)	11.4 ± 4.2 (8.8–14.1)	11.5 ± 4.3 (8.8–14.2)	0.57
Fat mass (%)	15.5 ± 3.9 (12.9–18.0)	15.6 ± 4.0 (13.0–18.2)	0.68
Fat-free mass (kg)	61.0 ± 6.1 (57.1–54.9)	60.9 ± 6.2 (57.0–64.9)	0.81
*Wingate performance*			
P. power (W)	744.7 ± 154.9 (646.2–843.1)	734.0 ± 132.4 (649.9–818.1)	0.55
P. power (W/kg)	10.1 ± 1.6 (9.1–11.1)	10.0 ± 0.98 (9.3–10.5)	0.59
M. power (W)	563.2 ± 101.9 (498.4–627.9)	572.3 ± 90.2 (514.9–629.5)	0.30
M. power (W/kg)	7.6 ± 0.9 (7.0–8.3)	7.7 ± 0.7 (7.3–8.2)	0.27
F. index (%)	52.8 ± 7.4 (48.1–57.5)	51.2 ± 6.9 (46.8–55.5)	0.66

Values are mean ± SD and 95% confidence interval. Abbreviations: P. power, peak power; M. power, mean power; F. index, fatigue index; T. work, total work.

**Table 3 ijerph-18-07227-t003:** Body composition and Wingate performance after 30 days of time-restricted feeding (TRF) and non-time-restricted feeding (non-TRF).

	TRF		Non-TRF	
	Before	30 days after	Before	30 days after
*Body composition*				
Body mass	73.6 ± 9.5 (67.5–79.5)	73.4 ± 9.3 (67.5–79.3)	73.5 ± 9.5 (67.5–79.5)	73.6 ± 9.5 (67.5–79.6)
Fat mass (kg)	11.4 ± 4.2 (8.8–14.1)	10.9 ± 3.9 (8.4–13.5)	11.6 ± 3.9 (9.1–14.1)	11.1 ± 3.9 (8.6–13.6)
Fat mass (%)	15.5 ± 3.9 (12.9–18.0)	14.8 ± 3.7 (12.4–17.2)	15.8 ± 3.5 (13.5–17.9)	15.1 ± 3.6 (12.7–17.4)
Fat-free mass (kg) *	61.0 ± 6.1 (57.1–54.9)	61.5 ± 6.3 (57.5–65.5)	60.8 ± 6.5 (56.7–64.9)	61.4 ± 6.6 (57.2–65.6)
*Wingate performance*				
P. power (W)	744.7 ± 154.9 (646.2–843.1)	742.5 ± 140.1 (653.5–831.5)	779.2 ± 143.8 (687.8–870.5)	761.1 ± 139.4 (672.5–849.6)
P. power (W/kg)	10.1 ± 1.6 (9.1–11.1)	10.1 ± 1.0 (9.4–10.7)	10.6 ± 1.3 (9.8–11.3)	10.3 ± 1.0 (9.6–10.9)
M. power (W) ^#^	563.2 ± 101.9 (498.4–627.9)	584.9 ± 101.8 (520.2–649.6)	579.5 ± 88.4 (523.3–635.7)	579.4 ± 83.8 (526.1–632.6)
M. power (W/kg)	7.6 ± 0.9 (7.0–8.3)	7.9 ± 0.7 (7.5–8.4)	7.9 ± 0.8 (7.4–8.3)	7.8 ± 0.6 (7.4–8.2)
F. index (%)	52.8 ± 7.4 (48.1–57.5)	52.4 ± 12.1 (44.7–60.1)	54.3 ± 10.5 (47.6–60.9)	51.4 ± 5.9 (47.5–55.1)
T. work (J) ^#^	16895.7 ± 3058.7 (14952.3–18839.0)	17547.5 ± 3056.0 (15605.7–19489.2)	17386.7 ± 2652.2 (15701.6–19071.9)	17382.1 ± 2513.2 (15785.2–18978.8)

Values are mean ± SD and 95% confidence interval. * Time main effect (*p* < 0.05), ^#^ time-by-condition interaction (*p* < 0.05). Abbreviations: P. power, peak power; M. power, mean power; F. index, fatigue index; T. work, total work.

## Data Availability

The datasets used and/or analyzed during the current study are available from the corresponding author on reasonable request.

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
