# Peer review of "Effects of Time-Restricted Feeding on Supramaximal Exercise Performance and Body Composition: A Randomized and Counterbalanced Crossover Study in Healthy Men"

_ijerph, 2021, doi:10.3390/ijerph18147227_

Round 1

Reviewer 1 Report

This paper describe using a crossover design a study assessing the effectiveness of short- and longer-term term restricted feeding on anaerobic performance measured using a 30-sec Wingate test.

Overall this is a nice tight study which whilst it has some limitations, t these are not fundamental and are recognised and acknowledged by the authors.

General Comments:

  • Was the diet prescribed
  • Ensure digit <10 are written as text not digits
  • In the main body avoid starting sentences with abbreviation s
  • I found the manuscript a little wordy in places I would ask the authors to read through and see where they might make it more concise.

Abstract:

  • Define TRF on first use

Method :

  • Consider adding body mass to the list of participant descriptive data.
  • What was the rationale for restricting to resistance trained individuals only, perhaps make clearer in the introduction.

Results:

  • I would suggest that you report Table 2 first as this confirms that your dietary intakes were similar.
  • As you are comparing response in both conditions one week after starting the study then as you do for table 3 report for both TRF and Non-TRF.
  • Table 3 caption suggests that you are reporting delta/change values which you are not so please amend wording.
  • Figure 2. please indicate what the relative values are relative to.

Author Response

Dear Reviewer

Please find attached our revision of manuscript ID: ijerph-1258740 “Effects of time-restricted feeding on supramaximal exercise performance and body composition: a randomized and counterbalanced crossover study in healthy men”. We thank the reviewers for their thoughtful comments that we believe have strengthened the manuscript. Below we describe in detail how we addressed each comment.

Thank you for your time and consideration.

Sincerely,

Joana M. Correia

Reviewer # 1:

This paper describe using a crossover design a study assessing the effectiveness of short- and longer-term term restricted feeding on anaerobic performance measured using a 30-sec Wingate test. Overall this is a nice tight study which whilst it has some limitations, t these are not fundamental and are recognised and acknowledged by the authors. 

General Comments:

  • Was the diet prescribed
  • Ensure digit <10 are written as text not digits
  • In the main body avoid starting sentences with abbreviation s
  • I found the manuscript a little wordy in places I would ask the authors to read through and see where they might make it more concise. 

Answer: Thank you for these observations. There was no diet prescription in what concerns to food type/portions. The dietary protocol (time-restricted feeding) was explained to all the participants, as described in the manuscript (page 4): “TRF interventions followed a 16/8 time-restricted feeding protocol [3]. All participants consumed two to three meals of ad libitum food intake during an 8-h period (between 1 and 9 pm). The remaining 16 hrs per 24-h time period constituted the fasting period during which participants only were allowed to consume water, tea and coffee (without caloric additives). The non-TRF diet corresponded to the participants’ usual dietary pattern without any timing restrictions”. We have followed the reviewers instructions regarding of ensuring that digits < 10 were written as text, avoiding starting sentences with abbreviations and reading carefully the entire manuscript, rephrasing some sentences.

Abstract:

  • Define TRF on first use 

Answer: We now defined TRF (time-restricted feeding).

Method :

  • Consider adding body mass to the list of participant descriptive data.

Answer: We now added mean body mass (page 2).

  • What was the rationale for restricting to resistance trained individuals only, perhaps make clearer in the introduction. 

Answer: Actually, we included individuals participating in power-sports training (which involve resistance training). We now added this information to the Methods section. Our rationale (as stated in the introduction) was that persons involved in power-sports modalities (e.g. sprint and middle-distance running) require high anaerobic capacity – namely, improved relative power in response to supramaximal exercise. Thus, we intended to explore whether TRF (a dietary approach that is well known to affect body composition) might improve WnT performance in persons involved in these sports. 

Results:

  • I would suggest that you report Table 2 first as this confirms that your dietary intakes were similar.

Answer: We proceeded accordingly.

  • As you are comparing response in both conditions one week after starting the study then as you do for table 3 report for both TRF and Non-TRF.

Answer: Measurements were taken before and after four weeks of each dietary intervention (TRF vs. non-TRF), as it can be seen in Figure 1. However, short-term effects (after one week) were only explored for the TRF protocol, since the literature shows that, physiological adaptations may occur after this specific time line (see for example, Klempel et al. 2010 Nutr J. 2010; 9: 35.).

  • Table 3 caption suggests that you are reporting delta/change values which you are not so please amend wording.

Answer: We agree with the reviewer and changed the title as suggested. It now reads “Table 3. Body composition and Wingate performance after 30 days of time restricted feeding (TRF) and non-time restricted feeding (non-TRF)”.

  • Figure 2. please indicate what the relative values are relative to.

Answer: Thank you for the suggestion. These values were calculated on an individual basis by dividing WnT absolute power by the participants' body mass. This information was added to the caption of fig. 2.

Reviewer 2 Report

  • The authors studied the effects of time-restricted feeding on supramaximal exercise performance and body composition: a randomized and counterbalanced crossover study in healthy men. This study is interesting, but there are minor issues that need to be fixed.
  • There are several grammatical issues throughout the text that need to be fixed.

Abstract

  • Line 1: If using an acronym, you must introduce it with complete terminology in the first instance, so your reader knows what it means.
  • The abstract needs to contain anthropometric characteristics of participants such as age, height, weight, etc.

Methods

  • Day-to-day test reliability, CV range, and intraclass correlation coefficients for the assessments need to be included for ALL the assessments.

Discussion

  • In this section, the authors need to explain the novelty of this study clearly

Author Response

Dear Reviewer

Please find attached our revision of manuscript ID: ijerph-1258740 “Effects of time-restricted feeding on supramaximal exercise performance and body composition: a randomized and counterbalanced crossover study in healthy men”. We thank the reviewers for their thoughtful comments that we believe have strengthened the manuscript. Below we describe in detail how we addressed each comment.

Thank you for your time and consideration.

Sincerely,

Joana M. Correia

Reviewer # 2:

The authors studied the effects of time-restricted feeding on supramaximal exercise performance and body composition: a randomized and counterbalanced crossover study in healthy men. This study is interesting, but there are minor issues that need to be fixed. There are several grammatical issues throughout the text that need to be fixed.

We reviewed the entire manuscript and hopefully have corrected all grammatical issues.

Abstract

  • Line 1: If using an acronym, you must introduce it with complete terminology in the first instance, so your reader knows what it means.

Answer: We agree with the reviewer and have now added ‘time-restricted feeding’.

  • The abstract needs to contain anthropometric characteristics of participants such as age, height, weight, etc.

Answer: We now added “(age: 22.4 ± 2.8 years, height: 174.0 ± 7.1 cm, body mass: 73.6 ± 9.5 kg, body mass index: 24.2 ± 2.0 kg/m2)”.

Methods

  • Day-to-day test reliability, CV range, and intraclass correlation coefficients for the assessments need to be included for ALL the assessments.

Answer: Thank you for this observation. We now added this information throughout the Methods section.

 (page 4) “Retest reliability and validity of power indices obtained during WnT testing have been well documented since the end of 1980 [10]. Nevertheless, more recent analyses revealed that the coefficient of variation for peak power, mean power, fatigue index and total work output is 0.97, 1.54, 1.89 and 1.74%, respectively. In addition, the intraclass correlation coefficients obtained for these respective variables was 0.987, 0.984, 0.948 and 0.968 [27]”.

(page 5) “Based on test-retest measures including 10 participants (other than the ones included in this study), the coefficient of variation for FM and FFM in our laboratory is 1.7 and 0.8%, respectively. The intraclass correlation coefficients obtained for these respective variables was 0.997 and 0.999 [28]”.

Discussion

  • In this section, the authors need to explain the novelty of this study clearly.

Answer: This is an important point. We added the following text to the beginning of the introduction:

“Our primary findings can be summarized as follows. First, contrasting to that typically reported after Ramadan, we found that WnT performance can be improved post-four weeks of TRF (vs. non-TRF) in healthy well-trained young men. Second, we also observed that short-term TRF is not an effective strategy for improving WnT performance and body composition in this specific population. Third, the magnitude of work-time improvement resulting from four weeks of TRF was greater than 1 s and this is quite meaningful from an athletic perspective.”

This manuscript is a resubmission of an earlier submission. The following is a list of the peer review reports and author responses from that submission.